# Safeguarding China's long-term sustainability against systemic disruptors

Ke Li [1,10], Lei Gao [2,10], Zhaoxia Guo [1,10], Yucheng Dong [1] ✉, Enayat A. Moallemi [3], Gang Kou [4,5], Meiqian Chen [1], Wenhao Lin[1], Qi Liu[1], Michael Obersteiner [6,7], Matteo Pedercini[8] & Brett A. Bryan [9]

China's long-term sustainability faces socioeconomic and environmental uncertainties. We identify five key systemic risk drivers, called *disruptors*, which could push China into a polycrisis: pandemic disease, ageing and shrinking population, deglobalization, climate change, and biodiversity loss. Using an integrated simulation model, we quantify the effects of these disruptors on the country's long-term sustainability framed by 17 Sustainable Development Goals (SDGs). Here we show that ageing and shrinking population, and climate change would be the two most influential disruptors on China's long-term sustainability. The compound effects of all disruptors could result in up to 2.1 and 7.0 points decline in the China's SDG score by 2030 and 2050, compared to the baseline with no disruptors and no additional sustainability policies. However, an integrated policy portfolio involving investment in education, healthcare, energy transition, water-use efficiency, ecological conservation and restoration could promote resilience against the compound effects and significantly improve China's long-term sustainability.

With the 2030 deadline for achieving the United Nations (UN)[1] 17 Sustainable Development Goals (SDGs) less than seven years away, robust actions are required to significantly improve the progress and accelerate the transition to a sustainable future. China has committed to meeting the SDGs and so far has taken important steps with crucial impacts at a global scale. For example, extreme poverty has been eradicated, and the targets of SDG 1.1 has been met 10 years ahead of schedule[2,3]. Long-term and large-scale investments in conservation and forest rehabilitation has enabled China to mitigate soil erosion and climate change, accounting for a quarter of the global newly increased vegetation areas from 2015 to 2020[4].

Despite important socioeconomic and environmental advances, China's progress towards the SDGs is still challenged by 2030 in the face of future irreducible uncertainties which we call disruptors. China

is confronted with multiple global disruptors such as infectious disease[5,6], deglobalization[7,8] and climate change[9,10] as well as by several domestic disruptors such as population shrinking and ageing[11,12] and ongoing biodiversity loss[13,14]. Understanding the effects of existing and future disruptors across scales is important for robust decision-making, from preparation to response and recovery.

At the global scale, few studies have analysed potential consequences of individual disruptors in isolation, such as economic and social impacts caused by the COVID-19 pandemic[15,16] or shocks to energy[17], water[18] and food systems[19] due to global supply chain disruptions during the Russo-Ukrainian War. However, multiple disruptors are complexly interconnected and are converging into a polycrisis[20,21], which further complicates the task of quantifying their effects on the SDGs. This necessitates a systemic exploration of their

[1]Business School, Sichuan University, Chengdu 610065, China. [2]Commonwealth Scientific and Industrial Research Organisation (CSIRO), Waite Campus, Adelaide, South Australia 5064, Australia. [3]Commonwealth Scientific and Industrial Research Organisation (CSIRO), Black Mountain, ACT, Australia. [4]Xiangjiang Laboratory, Changsha 410205, China. [5]School of Business Administration, Faculty of Business Administration, Southwestern University of Finance and Economics, Chengdu 610074, China. [6]International Institute for Applied Systems Analysis, Laxenburg 2361, Austria. [7]The Environmental Change Institute, University of Oxford, Oxford, UK. [8]Millennium Institute, Washington, DC, USA. [9]Centre for Integrative Ecology, School of Life and Environmental Sciences, Deakin University, Melbourne, Australia. [10]These authors contributed equally: Ke Li, Lei Gao, Zhaoxia Guo. ✉e-mail: ycdong@scu.edu.cn

**Table 1 | Overview descriptions of the five systemic disruptors**

| Disruptors | Descriptions |
|---|---|
| Pandemic disease | Emerging and re-emerging infectious disease pandemics, resulting in serious health crisis and economic recession. |
| Ageing and shrinking population | Demographic transition to an aging and shrinking population, which leads to shrinking labour force and requires more healthcare expenditure to support the needs of older adults. |
| Deglobalization | Deglobalization restricts access to foreign goods and services and adversely affects economic interactions between countries. Most countries would face high tariffs, financial constraints, and shrinking foreign trade and capital flows. |
| Climate change | Trade-offs between socioeconomic development and environmental impacts divert policy attention and resources away from climate action, resulting in higher energy and material consumption, slow progress on water-use efficiency, rising global temperatures, and an increase in the frequency and severity of extreme weather events. |
| Biodiversity loss | Lack of long-term attention and adequate financial support, combined with increasing resource demand and more environmental pollution associated with industrialisation and economic activities, leads to deforestation, terrestrial and marine ecosystem degradation, and reduced biodiversity. |

See Supplementary Table 1 for a detailed and quantitative description of each disruptor.

compound effects over time and across sectors. In China, previous studies have assessed SDG progress at the national[13,22] and local scales[23], identifying synergies and trade-offs[24,25], and analysing the effects of government policies[26,27]. However, these studies have been backwards-looking or assessed the status-quo of a few SDGs[23,25,28,29]. There is no forward-looking impact assessment of systemic disruptors on China's sustainability pointing to an important knowledge gap of disruptors' potential impact on the SDGs attainment and the role of robust response strategies.

In this study, we undertook a comprehensive assessment of China's future sustainable development subject to five major disruptors, including pandemic disease, ageing and shrinking population, deglobalization, climate change, and biodiversity loss (Table 1 and Supplementary Table 1). We quantified the potential consequences of China's five major disruptors for sustainable development, and assessed a range of policies for accelerating progress towards the SDGs. To assess the effects of these disruptors on China's sustainability for the years 2030 and 2050, we adopted a system dynamics model called the iSDG-China model (Methods). The iSDG-China model is based on the Integrated Sustainable Development Goal (iSDG) model[30] calibrated with historical data for China, and contains 30 interlinked modules distributed across all three pillars of sustainability (economy, society, and environment) and mapping key feedback loops and nonlinear relationships between and within these modules. Progress towards achieving the SDGs was quantified using the UN standard SDG score (i.e. 0 as the worst performance and 100 as full achievement), with individual and overall indices by 2030 and 2050 (Methods).

Using the model, we projected 2500 possibilities called states of the world under varying severity for one disruptor at a time, and 500 states of the world under the compound effects of interacting disruptors. We then constructed 243 sets of sectoral policies related to education, health, energy, water, and land called policy portfolios to mitigate the impacts of these disruptors and improve China's performance against the SDGs (Methods and Supplementary Table 2). We composed one baseline policy portfolio (no response), ten single-policy portfolios (policies in one sector at a time), and 232 integrated policy portfolios (policies in multiple sectors simultaneously). All policy portfolio and states of the world scenario combinations were compared to the Baseline scenario with no disruptors and no additional policy portfolios (Fig. 1).

## Results

### Ageing and shrinking population and climate change as China's most influential disruptors
The iSDG-China model projected that in a world free of disruptors and risk mitigation policy portfolios (Baseline), China's overall SDG score gradually improved from 68.3 points in 2022 to 71.0 and 72.5 points in 2030 and 2050 (Supplementary Fig. 1), respectively. This progress was mostly driven by improvements in food security (SDG 2), education

attainment (SDG 4), gender equality (SDG 5), access to clean water and sanitation (SDG 6), and economic growth (SDG 8). However, the Baseline scenario was far off full SDG attainment. Even slow progress might be achieved due to the presence of systemic disruptors, some of which China is already facing in reality. Our results suggest that each of the five disruptors undermined overall SDG performance (Fig. 2). Although the social and economic shocks triggered by the COVID-19 pandemic have seriously affected the implementation of the SDGs[15,16,31,32] in recent years, our results indicate that pandemic disease could be less disruptive to China's long-term sustainability than the emerging risk emanating from ageing and shrinking population, and climate change.

Under the scenarios of ageing and shrinking population, most SDGs were projected to perform worse, resulting in an average reduction of 0.3 and 1.5 points in the overall SDG score compared to the Baseline scenario in 2030 and 2050 (Fig. 2), respectively. By 2050, the working-age population (persons aged 15–65 years) decreased by about 53 million compared to the Baseline scenario (Supplementary Fig. 2). The Shrinking working-age population undermined economic development (SDG 8), with an average 3.6% increase in gross domestic product (GDP) per capita per year from 2022 to 2050, compared with the Baseline scenario average 5.1% growth per year (Supplementary Fig. 2). Based on the assumption that the structure of China's government expenditure remained unchanged compared to the Baseline scenario (Supplementary Table 3), the deteriorating economic outlook ultimately reduced government revenue (SDG 17) by 39.7% (Supplementary Fig. 2), and led to a decline in human well-being (SDGs 3 and 4) and biodiversity (SDGs 14 and 15), both heavily dependent on government expenditure. On the positive side, the slowdown of economic activity contributed to 15.4 and 39.8% reductions in water consumption (SDG 6) and greenhouse gas (GHG) emissions (SDG 13) by 2050 (Supplementary Fig. 2), compared to the Baseline scenario.

Climate change impacts, such as warming, water scarcity, and additional biodiversity loss, resulted in average reductions of 0.4 and 1.4 points in the overall SDG score compared to the Baseline scenario in 2030 and 2050 (Fig. 2). By 2050, under the climate change scenarios with the assumptions of lower energy and material efficiency, China involved a GHG emissions increase of 19.5% (SDG 13) on average compared to the Baseline scenario (Supplementary Fig. 2). In addition, slow improvements in water-use efficiency and accelerated warming contributed to a 23.2% increase in water consumption (SDG 6) (Supplementary Fig. 2), which also harmed life on land (SDG 15).

Furthermore, we calculated the main effects for each disruptor (Methods), which accounted for the uncertainty introduced by other disruptors when assessing the impacts of a specific disruptor on the SDGs. Based on these main effects, we identified ageing and shrinking population, and climate change as the two most influential disruptors affecting sustainability, with average reductions of 1.6 and 1.4 points in overall SDG scores by 2050 (Supplementary Fig. 3), respectively.

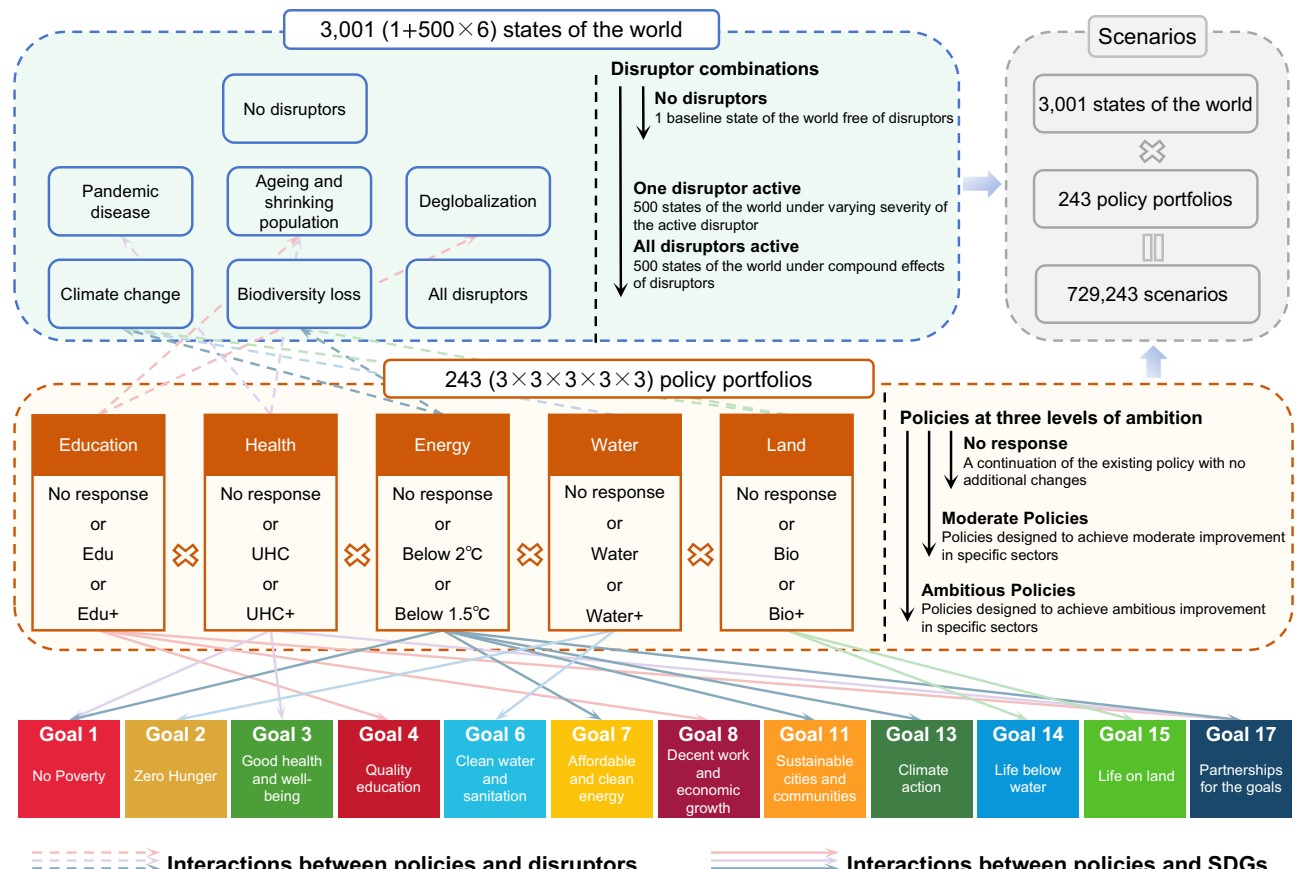

**Fig. 1 | Scenario design that combines states of the world with policy portfolios.**
First, states of the world were shaped by varying types and levels of the five disruptors. A total of 3001 states of the world involving no disruptors, only one disruptor, and all disruptors were considered. The baseline state of the world was a pathway along the current development trajectory and did not involve any disruptors. Then, five policy clusters related to a subset of the SDGs and disruptors were identified. Each policy cluster consisted of one 'no response' and two alternative policies to meet higher levels of improvements in specific sectors, including education, health, energy, water, and land. Each policy portfolio was formed by a combination of five policies from different policy clusters. By combining 3001 states of the world with 243 policy portfolios, we obtained a total of 729,243 scenarios. Each arrow between a policy and a disruptor or SDG indicates that the disruptor or SDG at the end point of the arrow is affected by the policy at the start point of the arrow.

## Concurrent disruptors lead to more severe impacts

The effects of concurrent disruptors could severely impact several SDGs, at a scale well beyond that of any one of these disruptors in isolation, suggesting complex system behaviour. The overall SDG score also experienced a more significant reduction due to the compound effects, with average reductions of 1.6 and 4.8 points by 2030 and 2050 (Fig. 2) compared to the Baseline scenario. We observed projected decreases in the overall SDG score of up to 2.1 and 7.0 points by 2030 and 2050 (Supplementary Fig. 4).

Such severe impacts were due to the compounding effects of interacting disruptors. For example, under the scenarios of all disruptors, additional public expenditure was required to support the health sector due to the ageing population and the reoccurrence of uncontrolled pandemic diseases. However, by 2050, all disruptors scenarios projected a 7.1% reduction in working-age population (Supplementary Fig. 2), and exports fell by 34.2% (Supplementary Fig. 2), which could undermine the potential of economic growth (SDG 8) and lead to a 52.3% reduction in government revenue, finally resulting in a more than doubling of the fiscal deficit (SDG 17) by 2050 (Supplementary Fig. 2). Widening deficits and increasing demand for health expenditure put good health and well-being (SDG 3) out of reach. In addition, a lack of adequate financial support for biodiversity conservation contributed to further biodiversity loss over the long term, resulting in reductions of 12.5 and 19.4 points in scores of SDGs 14 and 15 by 2050 (Fig. 2), compared to the Baseline scenario.

The compound effects of disruptors created dynamic knock-on effects by loss accumulation and feedback. By 2030, the scores of many SDGs were significantly reduced due to the compound effects, ranging from −0.1 to −7.1 points (Fig. 2) compared to the Baseline scenario. By 2050, the projected reductions significantly widened, ranging from −0.2 to −22.9 points (Fig. 2). The compound effects on the economy and society increased significantly over time. Compared to the Baseline scenario, the performance differences of no poverty (SDG 1), good health and well-being (SDG 3), decent work and economic growth (SDG 8) and partnerships (SDG 17) widened from −3.0, −3.2, −4.6 and −4.9 points by 2030 to −11.1, −10.6, −22.9 and −15.1 points by 2050 (Fig. 2). The compound effects on the environment were mixed: Climate action (SDG 13) performed 1.8 points better than the Baseline scenario by 2030 and 13.0 points better by 2050 (Fig. 2), while life below water (SDG 14) and life on land (SDG 15) performed worse, with the differences widening from −3.6 and −7.1 points in 2030 to −12.5 and −19.4 points in 2050 compared against the Baseline scenario (Fig. 2).

## Robust response to systemic disruptors for long-term sustainability

Our results show that diverse policy portfolios that cut across multiple sectors are required to mitigate systemic risk emanating from the compound effects of systemic disruptors on China's SDGs. Although all policy portfolios appeared superior in their overall SDG score

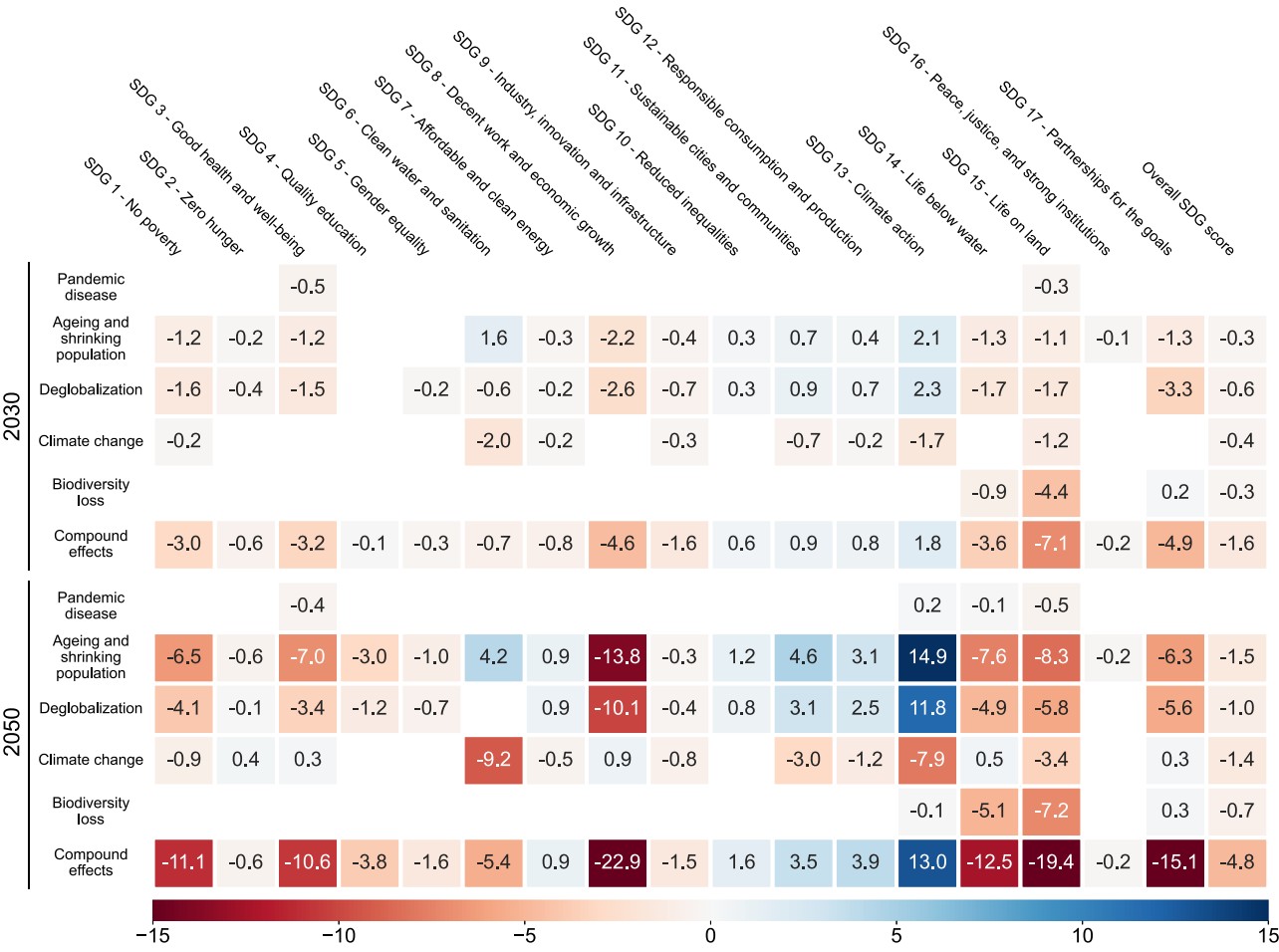

**Fig. 2 | The impacts of individual disruptors and the compound effects on the SDGs on average over the medium (2030) to long (2050) term.** The coloured shading indicates their performance differences as compared to the Baseline scenario in individual SDG scores and overall SDG scores. Cells are blank where the differences were small (i.e. between −0.1 and 0.1).

compared to the baseline policy portfolio, the improvements varied significantly, ranging from 0–2.6 points and 0.2–7.8 points by 2030 and 2050 under the states of the world involving no disruptors, respectively; while ranging from 0–2.2 points and 0.1–5.0 points by 2030 and 2050 under all disruptors (Fig. 3), respectively. This modelling exercise suggests that progress could mainly be driven by improvements in social and environmental sustainability. We projected the social and environmental SDG scores to range between 0 and 4.4 (0–6.2) points and 0–10.4 (0–13.4) points under the states of the world involving all disruptors (no disruptors) by 2050 (Fig. 3).

Single-policy portfolios exhibited considerable effectiveness in mitigating the negative effects of specific disruptors (Supplementary Figs. 5–9). Among these single-policy portfolios, environmental policies (i.e. energy, water and land) appeared to be most effective. For example, the below 2 °C and below 1.5 °C policies led to a marked increase in climate action (SDG 13), which was more significant under the climate change scenarios (Supplementary Fig. 8). The Bio and Bio+ policies contributed to considerable improvements in biodiversity (SDGs 14 and 15), especially under the biodiversity loss scenarios (Supplementary Fig. 9). Our simulations suggest that, across the scenarios, ambitious single-policy portfolios generated more synergies than moderate single-policy portfolios, with one exception (Supplementary Figs. 5–11). Comparing the below 2 °C to the below 1.5 °C policy impacts, we found that the additional energy system and carbon dioxide removal costs associated with the more ambitious climate policy brought trade-offs with social and economic sustainability, such

as higher government deficits (SDG 17), less funding for healthcare and education (SDGs 3 and 4), ultimately resulting in an inferior overall SDG score (Fig. 4 and Supplementary Fig. 12). Bridging financial gaps can address these trade-offs, e.g., carbon tax with revenue recycling can reduce emissions and improve well-being without undue burden on economic development[33].

Integrated policy portfolio simulations comprising policies from all five policy clusters performed better than single-policy portfolios. China would make the most significant progress under the Edu+, UHC +, Below 2 °C, Bio+ and Water+ policy portfolio, with the average overall SDG scores of 80 and 73 points under the states of the world involving no disruptors and all disruptors by 2050 (Fig. 3). Interestingly, the proactive Edu+, UHC+, Below 2 °C, Bio+, and Water+ policy portfolio simulation outperformed the progress towards sustainability under the Baseline scenario even considering the compound effects of disruptors. The implementation of the best-integrated policy portfolio could promote social and environmental sustainability, but could not fully insulate China from the economic consequences of the disruptors (Fig. 3), such as unsustainable debt (SDG 17) (Supplementary Figs. 13, 14).

## Discussion

The challenging prospect of achieving the SDGs by 2030 and beyond[34–37], further complicated by a series of possible disruptors[20,21], necessitates a comprehensive study to understand the interactions among various disruptors and their compound effects over time and

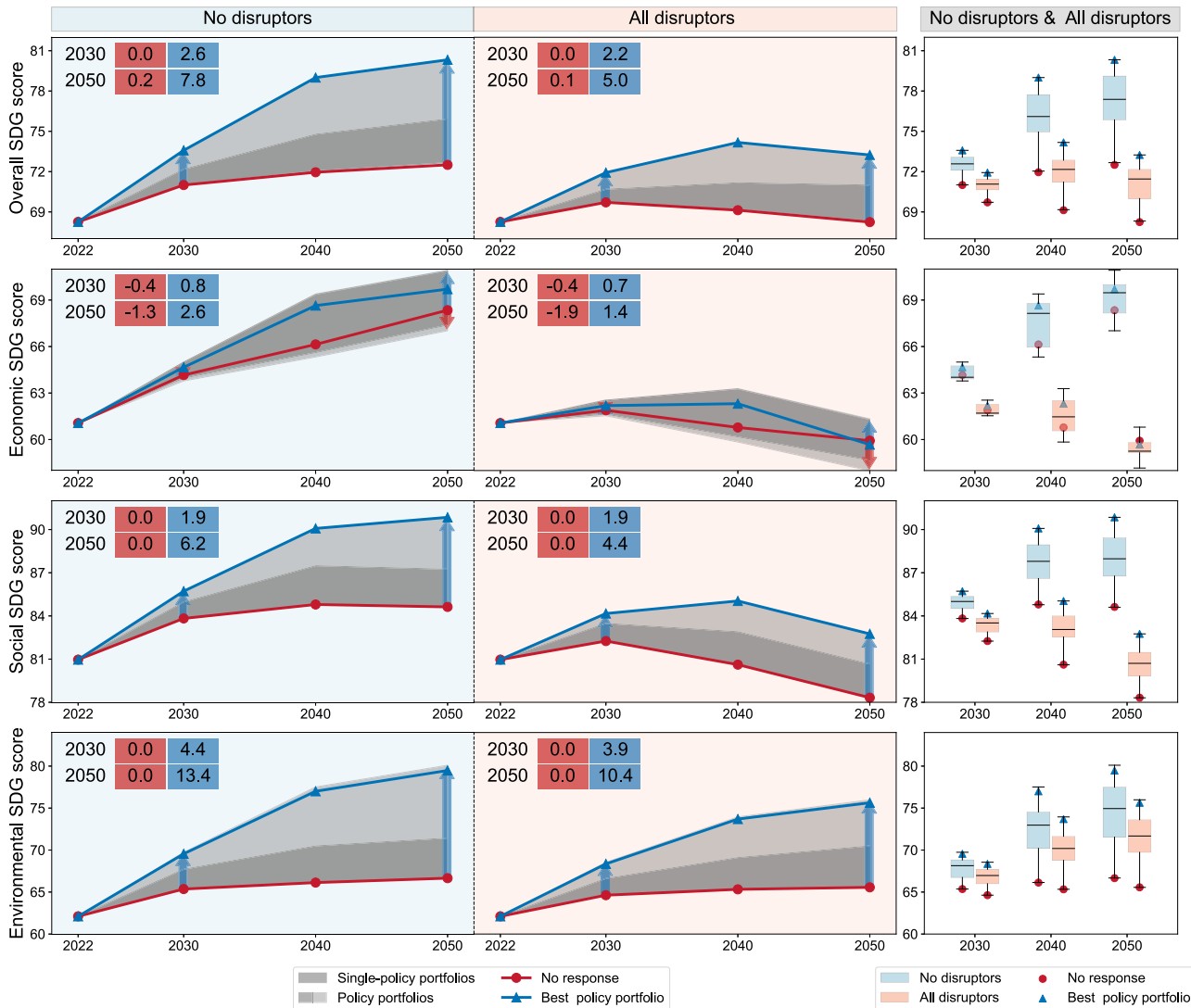

**Fig. 3 | The overall, economic, social and environmental SDG scores of the policy portfolios under two conditions: the states of the world without disruptors (in light blue) and states of the world with all disruptors (in light red) over time.** Supplementary Table 4 specifies SDG targets allocated to the economic, social and environmental categories, and the economic, social and environmental SDG scores are the mean of SDG targets for corresponding categories, respectively. The red line indicates the score of the baseline policy portfolio (no response) and the blue line represents the score of the best policy portfolio (Edu+, UHC+, Below 2 °C, Bio+ and Water+). The dark grey shading indicates the score range of single-policy portfolios over time. All shading (light grey shading and dark grey shading) indicates the score range of all policy portfolios over time. The numbers in coloured rectangles indicate the lower (in red) and upper bounds (in blue) of the score range of differences as compared against the baseline policy portfolio in the overall, economic, social and environmental SDG scores by 2030 and 2050. The boxplots indicate score distributions of the policy portfolios under the states of the world involving no disruptors (in light blue) and all disruptors (in light red) in 2030, 2040 and 2050, the red points indicate the scores of the baseline policy portfolio, and the blue triangles indicate the scores of the best policy portfolio.

across sectors. In response, our study quantified both the individual and compound effects of various disruptors on China's sustainability, including pandemic disease, ageing and shrinking population, deglobalization, climate change, and biodiversity loss; and explored integrated policy portfolios for safeguarding China's SDG implementation.

Most disruptors could potentially pose severe threats to China's long-term sustainability. Amongst the various disruptors, ageing and shrinking population, and climate change acted as the two most influential disruptors in the long term, resulting in reductions of 1.5 and 1.4 points of the overall SDG score by 2050, respectively. In line with previous research efforts[38–40] attempting to explore the implications of ageing on China, we found ageing and shrinking population posed significant challenges to the labour force vital for economic production, and placed unprecedented stress on the healthcare system, which would collectively reshape the trajectory of China's long-term economic

development, thereby severely affecting the progress of good health and well-being (SDG 3) and decent work and economic growth (SDG 8) (Fig. 2). Ageing and shrinking population also contributed to reductions in material consumption and emissions, affirming the findings from previous studies[41,42]. However, we found that it failed to deliver sustained and widespread environmental benefits, as evidenced by the continued poor performance in biodiversity (SDGs 14 and 15) under the assumption that government expenditure followed the baseline trend (Supplementary Table 3). In fact, the rising financial burden of ageing may force a reassessment of public spending priorities, potentially leading to reduced investments in land management, biodiversity conservation, and energy transition. Moreover, the preference of older adults for conventional products and services may weaken market demand for innovation and emerging technologies[43,44], potentially impeding the adoption of sustainable solutions such as the digital

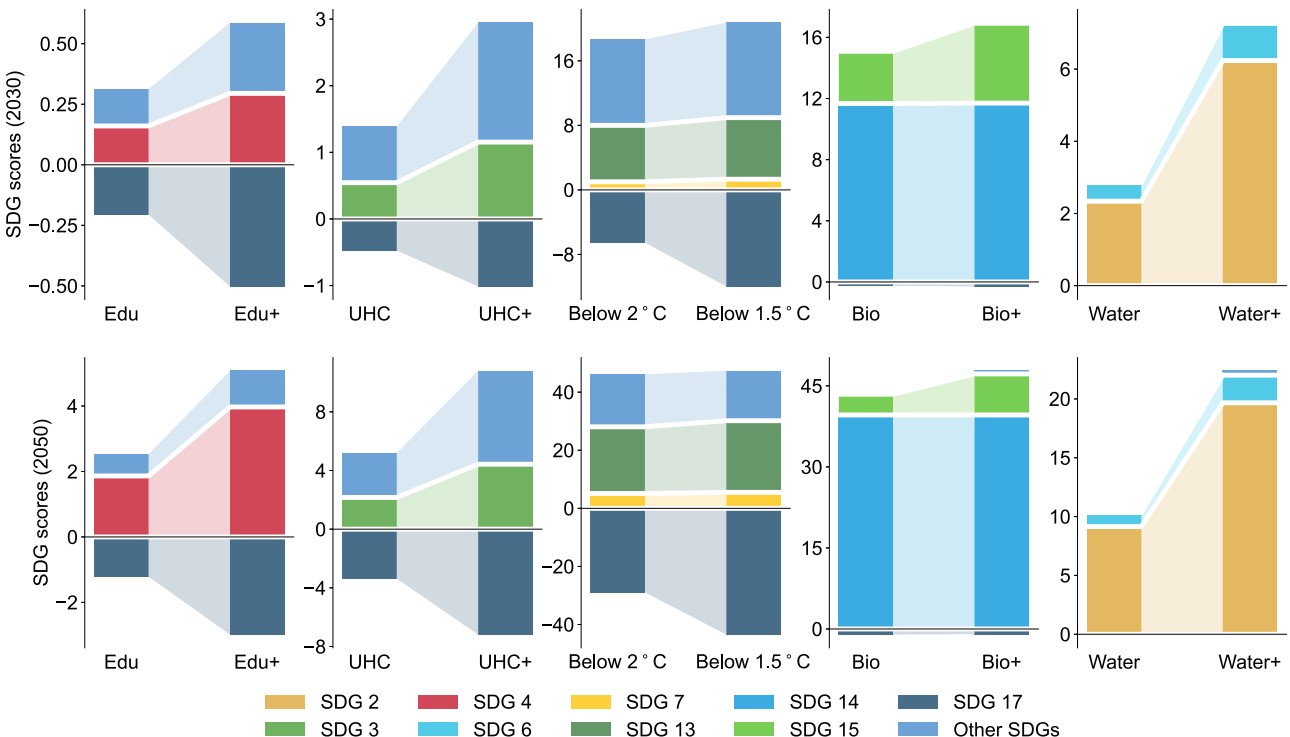

**Fig. 4 | Synergies and trade-offs of single-policy portfolios compared to the baseline policy portfolio under the states of the world with the compound effects from all disruptors by 2030 and 2050.** In each plot, each bar indicates single-policy portfolio performance differences as compared against the baseline policy portfolio in related SDGs by 2030 or 2050.

revolution and green consumption, which suggests that the real-world environmental impacts of ageing could be more negative.

There is growing evidence indicating that climate change poses significant challenges to the achievement of certain SDGs[45,46]. Our results show that, due to the nexus between climate change and sustainable development, the consequences of climate change were not limited to climate action (SDG 13) in China. The resulting cascades and feedbacks amplified the impacts of climate change on the environmental system and posed a severe threat to the attainment of other SDGs, including clean water and sanitation (SDG 6), sustainable cities and communities (SDG 11), and life on land (SDG 15). Such consequences amplified unevenness in China's SDG progress, as environmental sustainability lags behind in China (Supplementary Fig. 1). Therefore, as China endeavours to reach carbon-neutrality and achieve the SDGs simultaneously, forging connections between climate action and sustainable development is crucial to mitigate trade-offs and maximise synergies in achieving both climate and environmental SDGs, and inform policy decisions to build a future based on peace, stability, and shared prosperity.

Concurrent disruptors could have more widespread and far-reaching effects on SDG attainment. We projected that the achievement of most SDGs was threatened by the compound effects of five identified disruptors. While individual disruptors might occasionally bring about positive outcomes for some SDGs, these gains were often offset when multiple disruptors occurred simultaneously. Furthermore, a simultaneous occurrence of disruptors carried the potential for greater losses compared to the sum of the individual disruptors. This phenomenon reveals structural vulnerabilities of the whole system, highlighting the need to study the policies of achieving SDGs in the context of multiple disruptors. This was demonstrated during the COVID-19 pandemic, which initially threated public health, and then affected almost all aspects of the economy and society, and halted SDG progress in its tracks[15,32,47]. However, after the pandemic, the response to prevent the recurrence of similar events were far from sufficient[48]. In

a period of uncertainty defined by the ongoing polycrisis[21,49], limiting focus on only one subset of challenges or only a few SDGs could lead to an underestimation of the potential impacts of disruptors and unintended consequences.

Despite potentially facing multiple interacting disruptors, we found it is possible to promote resilience and accelerate progress towards the SDGs in China via the implementation of well-designed, integrated and proactive policy portfolios. Our simulations indicate that an integrated policy portfolio (Edu+, UHC+, Below 2 °C, Bio+, and Water+) in education, health, energy, water, and land sectors achieved more significant progress in SDG achievement over the long term. By 2050, compared to the baseline policy portfolio, the integrated policy portfolio achieved a 5-point improvement in the overall SDG score even considering the compound effects of all disruptors, and increased by 7.8 points on average when no disruptors were active. Although there exist differences in scenario design and indicator selection from recent studies[34,50] aimed at finding robust policies to facilitate sustainable development at a global scale, our study is consistent with these studies[34,50] in highlighting the importance of moving towards a future characterised by improved education, cleaner energy systems and more sustainable land use.

Although the integrated policy portfolio was effective, its implementation requires coordinated support and collaboration among relevant government agencies, and in reality, government agencies are separated and even sometimes compete with one another. Hence, genuine participation by a broad and diverse stakeholder group, including scientists, engineers, government, the corporate sector, and civil society must be encouraged to avoid the unintended consequences caused by a single-sector focus and to better benefit diverse groups[51]. Furthermore, cross-boundary information sharing and clear leadership are essential to coordinate the efforts of relevant government agencies. The national government can launch specific action plans to provide inter-departmental and provincial-level coordination mechanisms. Monitoring the quality of policy implementation and

strengthening mechanisms of accountability are also important for enduring sustainability outcomes. China's previous successes in handling complex issues via integrated policies, such as the programmes to improve land-system sustainability[26] and to eradicate extreme poverty[2], provide a proven template for addressing systemic disruptors and translating sustainability ambitions into meaningful action at the total system scale.

Additionally, policies aimed at addressing short-term needs when responding to a shock, need to align with the SDGs and other ambitious long-term national and international plans. However, there is some evidence that the implementation of long-term planning has been delayed by an over-prioritisation of current challenges and short-term decision-making. For example, China approved at least 50.4 gigawatts of coal power capacity in the first half of 2023[52], which aimed to recover the economy and enhance energy security, raising concerns about achieving China's carbon-neutrality target[53]. Increased investments in fossil energy and the delayed phase-out of fossil fuels[54,55] in the wake of the Russo-Ukrainian War has threatened climate-mitigation goals. It is hard to deal with concurrent, interacting, and compound disruptors by short-term decision-making, and this may even introduce new challenges. Thus, before the implementation of policies aimed at current challenges, our analyses highlight the need for thoughtful considerations of cascading effects on the whole system, and a careful balance between short-term interests and long-term benefits.

There exist some limitations to this study. First, due to the limited scope of the iSDG model, some domestic economic concerns, such as challenges in the property market and local government debts, as well as global economic challenges, such as the debt crises in developing countries and global economic slowdowns, have not been incorporated into our analyses, which have the potential to significantly influence China's economic trajectory and foreign economic cooperation over the medium to long term. Against the backdrop of escalating geopolitical tensions and global recession, these challenges could weaken China's adaptive capacity and further amplify the negative impacts of deglobalisation. Second, while the study included sectoral policies[50,56] related to education, health, energy, water, and land, more specific policies were not analysed in the study, including fertiliser consumption, dietary changes, carbon pricing policies, postponed retirement, as well as monetary and fiscal policies aimed at directly promoting economic development. Although these specific policies might yield influence on disruptors, integrating these policies needs a model capable of simulating the economy with enhanced precision and depth. Such complexity extends beyond the scope of the iSDG model, which is designed to capture broader interactions within the entire system. Future improvements in modelling capabilities could allow for a more detailed exploration of how these policies affect sustainability.

## Methods

### Overview of the iSDG model

The Integrated Sustainable Development Goals[30] (iSDG) model, developed by the Millennium Institute, is a system dynamics-based tool designed to explore and analyse medium- and long-term development issues. Within a single integrated framework, the iSDG model captures the complex interactions and feedback loops of economic, social, and environmental systems, and can provide a comprehensive assessment of progress towards the SDGs at the national level[57]. To cover relevant indicators for all 17 SDGs, the iSDG model includes over 3600 variables and several thousand feedback loops, and is composed of 30 modules, including ten social modules, ten economic modules and ten environmental modules. Each module interacts with other modules dynamically, forming a complex network of feedback loops that captures the development processes of the specific country. The overview of the iSDG model, the detailed description of each module

(e.g. module structure and major assumptions), and the interactions between modules are available in the iSDG documentation[30].

The iSDG model is based on the earlier Threshold 21 (T21) model which has improved over twenty years through implementation in over 40 countries, and leading to an integrated and proven modelling framework, with high flexibility and long-term scope[57]. These advantages make the iSDG model an ideal tool for assessing progress towards the SDGs at the national scale and support the simulation of various "what-if" scenarios[24]. Through a specific calibration and customisation process, the iSDG model can be customised for any country, and examples exist for multiple countries including Tanzania[58], Ivory Coast[59], and Australia[36]. In this study, we developed iSDG-China by customising the model based on China's specific conditions to simulate the progress on SDGs and support SDG planning.

### Model validation

Calibration of the iSDG-China model included both structural validation (to assess whether the concepts and interrelationships of the model conform to the corresponding knowledge from the real world) and behavioural validation (to compare simulation results with historical data), two techniques which are commonly used to validate system dynamics models[60]. The test of the model structure was the first step in validating the iSDG-China model. In the iSDG-China model, the syntax for the stock and flow diagram, feedback loops within and between modules, and the model equations and their reference parameter ranges were based on the expert reviews, other modelling studies, and empirical experience that emerged from the application of the T21 models, the earlier version of the iSDG model.

Behavioural validation was then used to examine how well-simulated results match the historical data of the real world using plotted graphs and goodness-of-fit metrics including the coefficient of determination ($R^2$), mean absolute error (MAE), mean squared error (MSE), root mean squared error (RMSE), and mean absolute percentage error (MAPE). The iSDG-China model was calibrated with historical time series data from 2000 to the most recent year for a selection of variables. Data were sourced primarily from official government sources, such as the China Statistical Yearbook and Finance Yearbook of China, as well as official data from international databases hosted by the International Monetary Fund, World Bank, and other official sources. The calibration results of simulations for a selection of variables are provided in Supplementary Fig. 15. We calculated goodness-of-fit metrics for a wider range of critical variables, which are shown in Supplementary Table 5. In addition, key indicators of the Baseline projections, such as demographics, economic growth, and fossil energy consumption, compared against projections from other relevant models for China are provided in Supplementary Figs. 16-18.

### Constructing the scenario framework

To provide insights into the robust assessment results of SDG progress, the impact of disruptors, and the effectiveness of policy portfolios in promoting long-term sustainability, we constructed a wide range of scenarios representing both socioeconomic development and policy possibilities of the future. The scenario framework construction was informed by previous studies[50,61,62], and each scenario was a combination of a state of the world and a policy portfolio. In this study, we developed 3,001 states of the world and 243 policy portfolios, which when combined together, formed 729,243 scenarios. The steps for constructing states of the world and policy portfolios are provided in the following subsection.

**States of the world.** States of the world aimed to describe a wide range of plausible trajectories of future socioeconomic development from an integrated perspective. We developed a baseline state of the world to project the future socioeconomic conditions following current trends up to 2050 based on a set of assumptions around

demographics, human development, economy and lifestyles, environment and resources, technology, and policy and institutions (Supplementary Table 3).

However, future socioeconomic and environmental conditions and sustainable development trajectories can be shaped by the types and severity of systemic disruptors. We first selected the main disruptors from the Global Risks Report[21] and scientific literature[63,64], based on three criteria. First, the disruptor should have a high likelihood of occurrence during a certain period. Second, the disruptor is likely to have a large and persistent impact on economic, social and environmental systems. Third, the disruptor should be able to be robustly quantified and incorporated into the iSDG-China model. As a result, we identified five disruptors, including pandemic disease, ageing and shrinking population, deglobalization, climate change, and biodiversity loss. A set of assumptions aligned with the performance of each disruptor were developed based on scientific literature and official time series datasets, which enable each disruptor to be parameterised into the iSDG-China model. The key parameters associated with the disruptors and their uncertainty ranges are presented in Supplementary Table 1.

We adopted the all-at-a-time method[65] (500 simulations, Latin hypercube sampling from uniform distribution) to perform Monte Carlo simulations for each disruptor individually and for all disruptors collectively. For each disruptor, we randomly sampled from the predetermined parameter ranges. When considering all disruptors, we randomly sampled from the combined parameter space of all disruptors. A total of 3000 alternative states of the world were generated, including 2500 states of the world under influenced solely by one disruptor, and 500 states of the world influenced by the compound effects of all disruptors simultaneously.

**Policy portfolios.** Using scientific literature[50,66], official data, government policy documents[67,68] and organisational reports[69], we identified five policy clusters of special importance for China's resilience against the disruptors and, more broadly, for sustainable development, including education, health, energy and climate, water and land. For each policy cluster, we developed policies at three levels of ambition: no response, moderate, and ambitious. The baseline policy from each policy cluster was a continuation of the existing policy with no additional changes while the moderate and ambitious policies were designed to achieve greater improvement in specific sectors. We developed 243 policy portfolios, with each policy portfolio consisting of five policies, one from each policy cluster. The baseline policy portfolio was comprised of five baseline policies, and single-policy portfolios included four baseline policies and only one moderate or ambitious policy. Detailed information on each policy is given in Supplementary Table 2.

### Assessing SDG implementation
We incorporated an evaluation framework into the iSDG-China model to assess progress towards the SDGs across different scenarios. To ensure comparability across indicators, the value of each indicator was rescaled from 0 (worst performance) to 100 (best performance). Our method for setting the upper and lower bounds followed the approach used in previous studies[13,22] and is shown below. Supplementary Table 4 presents the 87 SDG indicators covering all 17 SDGs, lower and upper indicator bounds, and the sources used to set them.

The upper bound for each indicator was determined using a five-step decision tree used in previous literature[13]. If the condition for a step is met, all of the later steps are skipped. First, we used relevant absolute quantitative thresholds for SDGs, such as "no poverty" or "absolute gender equality". Second, we adopted the UN Agenda 2030 core principle of leaving no one behind to determine the upper bound of zero deprivation or universal access for measures of extreme poverty, public service coverage, and access to basic infrastructure. Third,

we adopted bounds used in previous studies[13,22]. Fourth, we set the upper bound equal to the average of the top five performers (or the top three performers if fewer countries are included in the data). Fifth, all other indicators were set using a proportional improvement on a 2015 baseline value of 10–50%. The magnitude of the improvement was calibrated based on China's historic time series data and experience from other models.

The lower bound was defined using a three-step decision tree. First, we adopted the bounds used in the SDG Index and Dashboards Report[22]. Second, we set the lower bound equal to the values of the bottom 2.5th-percentile performer, which follows the criteria for selecting the lower bound in the SDG Index and Dashboards Report[22]. Third, for all other indicators, we used values of the worst performer from either China's historical time series data or from simulated data for 2000–2015.

After setting the lower and upper bounds for each, indicators were then linearly rescaled to values between 0 and 100. Each indicator was given a score of 100 if its value was greater than its corresponding upper bound, and a score of 0 if its value was less than its corresponding lower bound.

### Calculating the main effects of disruptors
In the design of experiments, a main effect is defined as the average effect of an independent variable on a dependent variable across different levels of other independent variables[70]. In this study, we used the main effects to quantify the average impacts of each disruptor on the SDGs across varying types and levels of other disruptors. Calculating the main effects of disruptors was limited to scenarios with no additional policies, and consisted of the processes of sampling and computation, as detailed below.

The sampling process consisted of two steps. In Step 1, we randomly set the state of each disruptor with 50% probability of being active. In Step 2, if a disruptor was active, we randomly sampled one from the predetermined ranges of the corresponding disruptor's parameters, representing the severity of the disruptor. Otherwise, we assigned the corresponding baseline value to the disruptor. Finally, Steps 1 and 2 were repeated 3000 times, resulting in 3000 states of the world. By combining the baseline policy portfolio, we then obtained 3000 scenarios.

We calculated main effects of five disruptors based on the 3000 scenarios. First, we ran model simulations under all scenarios. For each disruptor, we then categorised the model simulation results based on whether the disruptor was active or inactive under the corresponding scenarios. And the main effects of the disruptor could be calculated as the difference in the average SDG performance between two simulation categories.

We performed a calculation of the main effects of climate change as an example. First, we first randomly generated 3000 scenarios following the sampling process described above, and simulated SDG performance under each scenario. Then according to whether climate change was active or not in the scenarios, we divided the simulations into two groups, possibly resulting in about 1500 simulations with climate change and about 1500 simulations without climate change. The main effect of climate change could be obtained by calculating the difference between the average SDG performance of the two groups.

### Reporting summary
Further information on research design is available in the Nature Portfolio Reporting Summary linked to this article.

## Data availability
The dataset used for model calibration in the study is available in the public repository (https://doi.org/10.5281/zenodo.11443162). Additional materials and data are available in the Supplementary Information.

## Code availability

The iSDG model is owned by the Millennium Institute and is available from the Millennium Institute for research purposes on request. The codes used to produce the results of the study are available in the public repository (https://doi.org/10.5281/zenodo.11897708).

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

## Acknowledgements

This research was funded by the National Natural Science Foundation of China (No. 62350062, Y.D.; No. 72271171, Y.D.; No. 71910107002, G.K.), the Open Project of Xiangjiang Laboratory (No. 22XJ03028, Y.D.) and the Sichuan University (No. SKSYL2021-02, Y.D.).

## Author contributions

L.G., Z.G., Y.D., E.A.M. and B.A.B conceived and designed the study. M.P. developed the iSDG base model. K.L. and M.P. undertook model calibration for iSDG-China, scenario construction and simulations. K.L., L.G., Z.G. and Y.D. conducted the results analysis. K.L. created the figures. K.L., L.G., Z.G. and Y.D. prepared the manuscript. K.L., L.G., Z.G., Y.D., E.A.M., G.K., M.C, W.L., Q.L., M.O., M.P. and B.A.B participated in revising the manuscript, and all authors gave final approval for publication. Y.D., G.K., M.O. and B.A.B. supervised the work. K.L., L.G. and Z.G. contributed equally to the work.

## Competing interests

The authors declare no competing interests.
