## [Peer Review File · Nature Communications]

Safeguarding China's long-term sustainability against systemic disruptorsReviewers' Comments:

Reviewer #1:

Remarks to the Author:

This article is well written and makes transparent underlying models and assumptions, and the design of the simulation experiments on the system dynamic model used by the authors. The results enabling the authors to suggest certain policy directions are valuable, such as, "Our results show that diverse policy portfolios that cut across multiple sectors are required to mitigate systemic risk emanating from the compound effects of systemic disruptors on China's SDGs." And the details are in the article. Another finding advising against ambitious climate policy is, "Comparing 2°C to the 1.5°C policy impacts we found that the additional energy system and carbon dioxide removal costs associated with the more ambitious climate policy brought trade-offs with social and economic sustainability, such as higher government deficits (SDG 17), less funding for healthcare and education 217 (SDGs 3 and 4), ultimately resulting in an inferior overall SDG score."

This work would be significant if the authors could extend their discussion and show how this study has brought out insights and unique findings through simulation which were not predicted by other studies with similar concerns, that is, China's ability to achieve SDGs against disruptors. The impact of population aging on carbon emissions in China has been studied by others, (See <https://www.ncbi.nlm.nih.gov/pmc/articles/PMC9914734/> and other such articles.) The above-mentioned study brings out interesting findings relevant to what the authors are concerned about. Using Chinese provincial balanced panel data from 2000 to 2019 and extended Kaya and threshold effect models the study analyzes the impact of population aging on carbon emissions through consumption and production channels. The findings from this study could be compared by the authors, as the study concludes similarly, " Our results suggest that aging and shrinking population and climate change would be the two most influential systemic disruptors on China's long-term sustainability."

The authors must be appreciated for systematically explaining their model and its validation before using it for the simulation study. They have presented the study with useful graphics and conclusions. My concern is about the experimental design of the simulation experiments. The authors have clearly stated in the abstract of the paper, "China's long-term sustainability faces socioeconomic and environmental uncertainties. We identify five key systemic risk drivers, called disruptors, which could push China into a poly-crisis: pandemic disease, aging and shrinking population, deglobalization, climate change, and biodiversity loss."

Sure, as the authors claim, at the global scale, few studies have analyzed the potential consequences of individual disruptors in isolation.

So with this aim, the study should try to estimate the main effects of the five factors. Unfortunately, the experimental design given in Figure 1 (Scenario design that combines states of the world with policy portfolios), considers 3001 states of the world composed of 500 states for each of the individual disruptors, another 500 for all disruptors, and one state with no disruptors. A fractional factorial experiment involving all five factors to estimate the main effects and important interactions between factors with replications could have yielded a lot of useful data from the simulation experiments. The authors have used a full factorial experiment concerning the policy clusters. They are using $3 \times 3 \times 3 \times 3 \times 3 = 243$ experimental trials. (there is a typo $3 \times 3 \times 3 \times 3$ instead of $3 \times 3 \times 3 \times 3 \times 3$ appears.)

In my opinion, the authors have done one factor at a time experiment concerning disruptors. With a similar number of experimental trials much richer insight would have been possible. Also, the authors should explain how the 500 states of the world wrt to all disruptors are chosen.

The effect of the controllable factors(here, policy clusters) in reducing the harmful effect of the disruptors could be presented from such an experimental setup. Also, the authors could have analyzed the potential consequences of individual disruptors in isolation, from the main effects.

If the experimental efforts are not expensive this rectification could be attempted by the authors.

The value of the study will be enhanced if the revision

Reviewer #2:

Remarks to the Author:

By assessing the impact of 'systemic disruptors' on the SDG attainment and sustainability dynamics in China, the paper contributes to three fields: the existing literature that deals with SDG dynamics in China; the literature that uses/applies the iSDG model on specific countries; and a broader literature that uses integrated modeling and simulation techniques to explore how different policy responses may impact on SDG attainment dynamics.

The rationale/criteria for selecting 'systemic disruptors' are clearly presented, the iSDG model is adequately calibrated for the case of China, and overall the methodology of the research is clearly presented and effective.

The reported results are original (although not surprising) and can be used as a benchmark for future studies on China.

Despite its strengths, I think the paper in its current form has some notable weaknesses and would benefit from further work.

1. The 'discussion' section is rather weak. Here we would expect a more in-depth analysis on what we learn from the results/findings, e.g. on the channels, drivers, nodes, feedback loops between disruptors, response policies and SDGs dynamics, and how these findings relate to existing findings in the literature. In general, the paper is not very well-contextualised in the existing literature (e.g. shocks/crises and SDGs attainment, findings from other integrated simulation modelling on SDGs attainment in 2030/2050). Although this is not a significant weakness with regard to the paper's introduction, I think it is a weakness with regard to the paper's 'discussion/conclusion'.

2. Some aspects of the methodology/'research architecture' of the paper are not communicated very effectively. In particular the tripartite clustering of SDGs and disruptors (society, economy, environment), and the corresponding sectors involved, is not very clear (see: Table 1, analysis in lines 177-191, and Figure 1/sectors). For instance, it is not very clear to me why in Figure 1 deglobalisation/economy (together with ageing) is (only) under 'education'.

Further comments in case they are helpful to the authors.

3. The integration of the 'deglobalisation' as an economic disruptor in China is well-justified. Yet, the economic indicators included in "Table S1. Summary of main parameters and associated quantitative severity of the identified disruptors" may not fully capture the impact of 'deglobalisation' on the Chinese economy. Beyond the impact of rising geopolitical tensions and global fragmentation on the Chinese (export) economy (international level), the Chinese economy faces a number of domestic economic challenges / disruptions, e.g. challenges in the property market, domestic indebtedness (incl. local government), shadow economy and private sector exposure] that seems to be left outside from the analysis. Modelling for different 'economic stress' scenarios would be interesting/important here (one could include here the impact of debt crises in developing countries across the BRI; and scenarios for global economic slowdowns?). Finally, it is not very clear to me how 'economic growth' projections are integrated in the medium/long term simulations. Historical trends (as presented in "Supplementary Figures") may not be a good guide for global or Chinese economic growth dynamics in the next 7-20 years.

Brief comments:

- i. The difference of the impact between 'ageing only' (-39.7%) and 'all disruptors' (50%) scenarios on 'government revenue' seems rather small?
- ii. The differential impact that disruptors (e.g. 'ageing') has on environmental variables (biodiversity vs. CC) could be discussed a bit more (see also the respective literature on economic/financial crises and economic slowdowns).
- iii. The different 'levels of ambition' in policy responses/portfolios are not clearly introduced in the paper. Lines 404-416 refer to three levels: no response, moderate, ambitious. Yet, the paper reports only on 'no response' and 'ambitious'? A clearer presentation of this in the paper would make a difference.
- iv. There seems to be a mismatch between the presentation in Figure 3 and corresponding lines 233-245 (maybe the problem is with the headings 'no disruptor', 'all disruptors?'). Please check. Also, the findings reported here are not very clear to me. Is it that in an 'all disruptors' & 'no response' scenario the impact on SDG attainment until 2030 is negligible? Is this possible?
- v. In Figure 2: why there are no data for some variables?
- vi. In 'Extended Data Figure 6'. Check the data for Government revenue. Why does it increase with Climate Change? Am I misreading this?

February 29, 2024

Re: Response to Reviewer comments

We thank the reviewers for their insightful comments and valuable suggestions, which guide us to enhance the quality of our manuscript. In response, we made three major revisions to the manuscript: (1) we revised the discussion section to highlight our unique contributions on China's SDG attainment; (2) we added the main effects of individual disruptors to enhance the robustness of our conclusions; and (3) we revised Figures 1 and 3 to clearly illustrate the scenario construction and our results. In the following, we provided a point-by-point response (in blue) to both reviewers. *“Blue italics in double quotes are texts in the revised manuscript”*.

Reviewer #1's Comments:

This article is well written and makes transparent underlying models and assumptions, and the design of the simulation experiments on the system dynamic model used by the authors. The results enabling the authors to suggest certain policy directions are valuable, such as, “Our results show that diverse policy portfolios that cut across multiple sectors are required to mitigate systemic risk emanating from the compound effects of systemic disruptors on China's SDGs.” And the details are in the article. Another finding advising against ambitious climate policy is, “Comparing 2°C to the 1.5°C policy impacts we found that the additional energy system and carbon dioxide removal costs associated with the more ambitious climate policy brought trade-offs with social and economic sustainability, such as higher government deficits (SDG 17), less funding for healthcare and education (SDGs 3 and 4), ultimately resulting in an inferior overall SDG score.”

Authors' response: Thank you very much for your positive comments. Following the comments, we made substantial improvements in the revised manuscript. Below is our detailed response to the comments, and we hope the revisions address your concerns.

Comment 1: This work would be significant if the authors could extend their discussion and show how this study has brought out insights and unique findings through simulation which were not predicted by other studies with similar concerns, that is, China's ability to achieve SDGs against disruptors. The impact of population aging on carbon emissions in China has been studied by others, (See <https://www.ncbi.nlm.nih.gov/pmc/articles/PMC9914734/> and other such articles.) The above-mentioned study brings out interesting findings relevant to what the authors are concerned about. Using Chinese provincial balanced panel data from 2000 to 2019

and extended Kaya and threshold effect models the study analyzes the impact of population aging on carbon emissions through consumption and production channels. The findings from this study could be compared by the authors, as the study concludes similarly, “Our results suggest that aging and shrinking population and climate change would be the two most influential systemic disruptors on China’s long-term sustainability.”

Authors’ response: Thank you for the insightful comment. In the revised manuscript, we clarified our unique contributions on China’s SDG attainment. Please see Page 9 of the revised manuscript.

“The challenging prospect of achieving the SDGs by 2030 and beyond^{33–36}, further complicated by a series of possible disruptors^{20,21}, necessitates a comprehensive study to understand the interactions among various disruptors and their compound effects over time and across sectors. In response, our study quantified both the individual and compound effects of various disruptors on China’s sustainability, including pandemic disease, ageing and shrinking population, deglobalization, climate change, and biodiversity loss; and explored integrated policy portfolios for safeguarding China’s SDG implementation.”

In addition, starting from the key reference you provided, we extended the discussion of ageing and shrinking population and climate change, the two most influential disruptors affecting China’s SDGs, by comparing our findings with those of existing relevant studies. Please see Page 10 of the revised manuscript.

“In line with previous research efforts^{37–39} attempting to explore the implications of ageing on China, we found ageing and shrinking population posed significant challenges to the labour force vital for economic production, and placed unprecedented stress on the healthcare system, which would collectively reshape the trajectory of China’s long-term economic development, thereby severely affecting the progress of good health and well-being (SDG 3) and decent work and economic growth (SDG 8) (Fig. 2). Ageing and shrinking population also contributed to reductions in material consumption and emissions, affirming the findings from previous studies^{40,41}. However, we found that it failed to deliver sustained and widespread environmental benefits, as evidenced by the continued poor performance in biodiversity (SDGs 14 and 15) under the assumption that government expenditure followed the baseline trend (Supplementary Table 3). In fact, the rising financial burden of ageing may force a reassessment of public spending priorities, potentially leading to reduced investments in land management, biodiversity conservation, and energy transition. Moreover, the preference of older adults for conventional products and services may weaken market demand for innovation and emerging technologies^{42,43}, potentially impeding the adoption of sustainable solutions such as the digital revolution and green consumption, which suggests that the real-world environmental impacts of ageing could be more negative.”

“There is growing evidence indicating that climate change poses significant

challenges to the achievement of certain SDGs^{44,45}. Our results show that, due to the nexus between climate change and sustainable development, the consequences of climate change were not limited to climate action (SDG 13) in China. The resulting cascades and feedbacks amplified the impacts of climate change on the environmental system and posed a severe threat to the attainment of other SDGs, including clean water and sanitation (SDG 6), sustainable cities and communities (SDG 11), and life on land (SDG 15). Such consequences amplified unevenness in China's SDG progress, as environmental sustainability lags behind in China (Supplementary Fig. 1). Therefore, as China endeavours to reach carbon-neutrality and achieve the SDGs simultaneously, forging connections between climate action and sustainable development is crucial to mitigate trade-offs and maximize synergies in achieving both climate and environmental SDGs, and inform policy decisions to build a future based on peace, stability, and shared prosperity."

Comment 2: The authors must be appreciated for systematically explaining their model and its validation before using it for the simulation study. They have presented the study with useful graphics and conclusions. My concern is about the experimental design of the simulation experiments. The authors have clearly stated in the abstract of the paper, "China's long-term sustainability faces socioeconomic and environmental uncertainties. We identify five key systemic risk drivers, called disruptors, which could push China into a poly-crisis: pandemic disease, aging and shrinking population, deglobalization, climate change, and biodiversity loss." Sure, as the authors claim, at the global scale, few studies have analyzed the potential consequences of individual disruptors in isolation. So with this aim, the study should try to estimate the main effects of the five factors. Unfortunately, the experimental design given in Figure 1 (Scenario design that combines states of the world with policy portfolios), considers 3001 states of the world composed of 500 states for each of the individual disruptors, another 500 for all disruptors, and one state with no disruptors. A fractional factorial experiment involving all five factors to estimate the main effects and important interactions between factors with replications could have yielded a lot of useful data from the simulation experiments. The authors have used a full factorial experiment concerning the policy clusters. They are using $3 \times 3 \times 3 \times 3 \times 3 = 243$ experimental trials.

In my opinion, the authors have done one factor at a time experiment concerning disruptors. With a similar number of experimental trials much richer insight would have been possible. The effect of the controllable factors (here, policy clusters) in reducing the harmful effect of the disruptors could be presented from such an experimental setup. Also, the authors could have analyzed the potential consequences of individual disruptors in isolation, from the main effects.

If the experimental efforts are not expensive this rectification could be attempted by the authors. The value of the study will be enhanced if the revision.

Authors' response: Thank you for the insightful comment. We fully agree with this comment. In the revised manuscript, we tried to balance the complexity of experiments with the robustness of our results. Specifically, we took the following two actions:

(1) We added the text in the main text (see Page 5) and Supplementary Fig. 3 (see below) to analyse the main effects of each disruptor.

“Furthermore, we calculated the main effects for each disruptor (Methods), which accounted for the uncertainty introduced by other disruptors when assessing the impacts of a specific disruptor on the SDGs. Based on these main effects, we identified ageing and shrinking population, and climate change as the two most influential disruptors affecting sustainability, with average reductions of 1.6 and 1.4 points in overall SDG scores by 2050 (Supplementary Fig. 3), respectively.”

Supplementary Figure 3. The main effects of individual disruptors on the SDGs over the medium (2030) to long (2050) term. The coloured shading indicates main effects of individual disruptors on individual SDG scores and overall SDG scores. See Methods for the detailed calculation process of main effects. Cells are blank where the differences were small (i.e., between -0.1 and 0.1).

(2) We added the text in the main text (see Page 7) and Supplementary Figs. 5-9 in the Supplementary Information to analyze the roles of single-policy portfolios in reducing the negative effects of specific disruptors.

“Single-policy portfolios exhibited considerable effectiveness in mitigating the negative effects of specific disruptors (Supplementary Figs. 5-9). Among these single-policy portfolios, environmental policies (i.e., energy, water, and land) appeared to be most effective. For example, the below 2°C and below 1.5°C policies led to a marked increase in climate action (SDG 13), which was more significant under the climate change scenarios (Supplementary Fig. 8). The Bio and Bio+ policies contributed to considerable improvements in biodiversity (SDGs 14 and 15), especially under the biodiversity loss scenarios (Supplementary Fig. 9).”

Comment 3: Also, the authors should explain how the 500 states of the world wrt to all disruptors are chosen.

Authors’ response: Apologies for the confusion. We revised the related text in

Methods sub-section of the revised manuscript to clarify this issue.

“We adopted the all-at-a-time method⁶⁴ (500 simulations, Latin hypercube sampling from uniform distribution) to perform Monte Carlo simulations for each disruptor individually and for all disruptors collectively. For each disruptor, we randomly sampled from the predetermined parameter ranges. When considering all disruptors, we randomly sampled from the combined parameter space of all disruptors. A total of 3,000 alternative states of the world were generated, including 2500 states of the world under influenced solely by one disruptor, and 500 states of the world influenced by the compound effects of all disruptors simultaneously.”

Moreover, we revised Fig. 1 to better illustrate the scenario construction.

Figure 1. Scenario design that combines states of the world with policy portfolios. First, states of the world were shaped by varying types and levels of the five disruptors. A total of 3,001 states of the world involving no disruptors, only one disruptor, and all disruptors were considered. The baseline state of the world was a pathway along the current development trajectory and did not involve any disruptors. Then, five policy clusters related to a subset of the SDGs and disruptors were identified. Each policy cluster consisted of one “no response” and two alternative policies to meet higher levels of improvements in specific sectors, including education, health, energy, water, and land. Each policy portfolio was formed by a combination of five policies from different policy clusters. By combining 3,001 states of the world with 243 policy portfolios, we obtained a total of 729,253 scenarios. Each arrow between a policy and a disruptor or SDG indicates that the disruptor or SDG at the end point of the arrow is affected by the policy at the start point of the arrow.

Comment 4: There is a typo 3x3x3x3 instead of 3x3x3x3x3 appears.

Authors’ response: Fixed. Thank you!

Thank you again for your constructive comments and support! Your comments and recommendations have aided us in strengthening this work.

Reviewer #2's Comments:

By assessing the impact of 'systemic disruptors' on the SDG attainment and sustainability dynamics in China, the paper contributes to three fields: the existing literature that deals with SDG dynamics in China; the literature that uses/applies the iSDG model on specific countries; and a broader literature that uses integrated modeling and simulation techniques to explore how different policy responses may impact on SDG attainment dynamics. The rationale/criteria for selecting 'systemic disruptors' are clearly presented, the iSDG model is adequately calibrated for the case of China, and overall the methodology of the research is clearly presented and effective. The reported results are original (although not surprising) and can be used as a benchmark for future studies on China.

Authors' response: We highly appreciate your thoughtful comments, and they have aided us in strengthening this manuscript. Below we carefully addressed all comments from you.

Comment 1: Despite its strengths, I think the paper in its current form has some notable weaknesses and would benefit from further work. The 'discussion' section is rather weak. Here we would expect a more in-depth analysis on what we learn from the results/findings, e.g. on the channels, drivers, nodes, feedback loops between disruptors, response policies and SDGs dynamics, and how these findings relate to existing findings in the literature. In general, the paper is not very well-contextualised in the existing literature (e.g. shocks/crises and SDGs attainment, findings from other integrated simulation modelling on SDGs attainment in 2030/2050). Although this is not a significant weakness with regard to the paper's introduction, I think it is a weakness with regard to the paper's 'discussion/conclusion'.

Authors' response: Thank you for the insightful comment. Following this comment, we rewrote the discussion section (see Pages 9-12) to highlight our findings and to strengthen the comparisons with the existing findings in the literature. Specifically, we took the following actions:

(1) We clarified our unique contributions on China's SDG attainment.

"The challenging prospect of achieving the SDGs by 2030 and beyond³³⁻³⁶, further complicated by a series of possible disruptors^{20,21}, necessitates a comprehensive study to understand the interactions among various disruptors and their compound effects over time and across sectors. In response, our study quantified both the individual and compound effects of various disruptors on China's sustainability, including pandemic disease, ageing and shrinking population, deglobalization, climate change, and biodiversity loss; and explored integrated policy portfolios for safeguarding China's SDG implementation."

(2) We extended the discussion of ageing and shrinking population and climate change.

"Most disruptors could potentially pose severe threats to China's long-term sustainability. Amongst the various disruptors, ageing and shrinking population, and climate change acted as the two most influential disruptors in the long term,

resulting in reductions of 1.5 and 1.4 points of the overall SDG score by 2050, respectively. In line with previous research efforts^{37–39} attempting to explore the implications of ageing on China, we found ageing and shrinking population posed significant challenges to the labour force vital for economic production, and placed unprecedented stress on the healthcare system, which would collectively reshape the trajectory of China's long-term economic development, thereby severely affecting the progress of good health and well-being (SDG 3) and decent work and economic growth (SDG 8) (Fig. 2). Ageing and shrinking population also contributed to reductions in material consumption and emissions, affirming the findings from previous studies^{40,41}. However, we found that it failed to deliver sustained and widespread environmental benefits, as evidenced by the continued poor performance in biodiversity (SDGs 14 and 15) under the assumption that government expenditure followed the baseline trend (Supplementary Table 3). In fact, the rising financial burden of ageing may force a reassessment of public spending priorities, potentially leading to reduced investments in land management, biodiversity conservation, and energy transition. Moreover, the preference of older adults for conventional products and services may weaken market demand for innovation and emerging technologies^{42,43}, potentially impeding the adoption of sustainable solutions such as the digital revolution and green consumption, which suggests that the real-world environmental impacts of ageing could be more negative.”

“There is growing evidence indicating that climate change poses significant challenges to the achievement of certain SDGs^{44,45}. Our results show that, due to the nexus between climate change and sustainable development, the consequences of climate change were not limited to climate action (SDG 13) in China. The resulting cascades and feedbacks amplified the impacts of climate change on the environmental system and posed a severe threat to the attainment of other SDGs, including clean water and sanitation (SDG 6), sustainable cities and communities (SDG 11), and life on land (SDG 15). Such consequences amplified unevenness in China's SDG progress, as environmental sustainability lags behind in China (Supplementary Fig. 1). Therefore, as China endeavours to reach carbon-neutrality and achieve the SDGs simultaneously, forging connections between climate action and sustainable development is crucial to mitigate trade-offs and maximize synergies in achieving both climate and environmental SDGs, and inform policy decisions to build a future based on peace, stability, and shared prosperity.”

(3) We highlighted the compound effects of all disruptors on SDG attainment.

“Concurrent disruptors could have more widespread and far-reaching effects on SDG attainment. We projected that the achievement of most SDGs was threatened by the compound effects of five identified disruptors. While individual disruptors might occasionally bring about positive outcomes for some SDGs, these gains were often offset when multiple disruptors occurred simultaneously. Furthermore, a simultaneous occurrence of disruptors carried the potential for greater losses

compared to the sum of the individual disruptors. This phenomenon reveals structural vulnerabilities of the whole system, highlighting the need to study the policies of achieving SDGs in the context of multiple disruptors. This was demonstrated during the COVID-19 pandemic, which initially threatened public health, and then affected almost all aspects of the economy and society, and halted SDG progress in its tracks^{15,32,46}. However, after the pandemic, the response to prevent the recurrence of similar events were far from sufficient⁴⁷. In a period of uncertainty defined by the ongoing polycrisis^{21,48}, limiting focus on only one subset of challenges or only a few SDGs could lead to an underestimation of the potential impacts of disruptors and unintended consequences.”

(4) We added the text to discuss the best policy portfolio and its implementation.

“Despite potentially facing multiple interacting disruptors, we found it is possible to promote resilience and accelerate progress towards the SDGs in China via the implementation of well-designed, integrated and pro-active policy portfolios. Our simulations indicate that an integrated policy portfolio (Edu+, UHC+, Below 2°C, Bio+ and Water+) in education, health, energy, water, and land sectors achieved more significant progress in SDG achievement over the long term. By 2050, compared to the baseline policy portfolio, the integrated policy portfolio achieved a 5-point improvement in the overall SDG score even considering the compound effects of all disruptors, and increased by 7.8 points on average when no disruptors were active. Although there exist differences in scenario design and indicator selection from recent studies^{33,49} aimed at finding robust policies to facilitate sustainable development at a global scale, our study is consistent with these studies^{33,49} in highlighting the importance to move towards a future characterised by improved education, cleaner energy systems and more sustainable land use ...”

Comment 2: Some aspects of the methodology/’research architecture’ of the paper are not communicated very effectively. In particular the tripartite clustering of SDGs and disruptors (society, economy, environment), and the corresponding sectors involved, is not very clear (see: Table 1, analysis in lines 177-191, and Figure 1/sectors). For instance, it is not very clear to me why in Figure 1 deglobalisation/economy (together with ageing) is (only) under ‘education’.

Authors’ response: We sincerely apologize for any confusion caused by the presentation of the methodology of our study. In the revised manuscript, we revised Fig. 1 to provide a clearer explanation for each cluster of SDGs and disruptors.

Figure 1. Scenario design that combines states of the world with policy portfolios. First, states of the world were shaped by varying types and levels of the five disruptors. A total of 3,001 states of the world involving no disruptors, only one disruptor, and all disruptors were considered. The baseline state of the world was a pathway along the current development trajectory and did not involve any disruptors. Then, five policy clusters related to a subset of the SDGs and disruptors were identified. Each policy cluster consisted of one “no response” and two alternative policies to meet higher levels of improvements in specific sectors, including education, health, energy, water, and land. Each policy portfolio was formed by a combination of five policies from different policy clusters. By combining 3,001 states of the world with 243 policy portfolios, we obtained a total of 729,253 scenarios. Each arrow between a policy and a disruptor or SDG indicates that the disruptor or SDG at the end point of the arrow is affected by the policy at the start point of the arrow.

Meanwhile, in the revised Table 1, we removed the classification of disruptors to avoid confusions. In addition, we revised text in Lines 177-191.

Table 1. Overview descriptions of the five systemic disruptors. See Supplementary Table 1 for a detailed and quantitative description of these disruptors.

Disruptors	Descriptions
Pandemic disease	Emerging and re-emerging infectious disease pandemics, resulting in serious health crisis and economic recession.
Ageing and shrinking population	Demographic transition to an aging and shrinking population, which leads to shrinking labour force and requires more healthcare expenditure to support the needs of older adults.
Deglobalization	Deglobalization restricts access to foreign goods and services and adversely affects economic interactions between countries. Most countries would face high tariff, financial constraints, and shrinking foreign trade and capital flows.
Climate change	Trade-offs between socioeconomic development and environmental impacts diverts policy attention and resources away from climate action, resulting in higher energy and material consumption, slow progress on water-use efficiency, rising global temperatures, and an increase in the frequency and severity of extreme weather events.

Biodiversity loss

Lack of long-term attention and adequate financial support, combined with increasing resource demand and more environmental pollution associated with industrialization and economic activities, leads to deforestation, terrestrial and marine ecosystem degradation, and reduced biodiversity.

Comment 3: Further comments in case they are helpful to the authors. The integration of the ‘deglobalisation’ as an economic disruptor in China is well-justified. Yet, the economic indicators included in “Table S1. Summary of main parameters and associated quantitative severity of the identified disruptors” may not fully capture the impact of ‘deglobalisation’ on the Chinese economy. Beyond the impact of rising geopolitical tensions and global fragmentation on the Chinese (export) economy (international level), the Chinese economy faces a number of domestic economic challenges / disruptions, e.g. challenges in the property market, domestic indebtedness (incl. local government), shadow economy and private sector exposure] that seems to be left outside from the analysis. Modelling for different ‘economic stress’ scenarios would be interesting/important here (one could include here the impact of debt crises in developing countries across the BRI; and scenarios for global economic slowdowns?).

Authors’ response: We fully agree your comment. Incorporating the domestic and global economic challenges could provide a more comprehensive assessment of the effects of deglobalization within the Chinese economy. However, it is not yet possible for the iSDG model to simulate these challenges. We clarified the limitation within the discussion section of our revised manuscript. Please see Page 12 of the revised manuscript.

“There exist some limitations to this study. First, due to the limited scope of the iSDG model, some domestic economic concerns, such as challenges in the property market and local government debts, as well as global economic challenges, such as the debt crises in developing countries and global economic slowdowns, have not been incorporated into our analyses, which have the potential to significantly influence China’s economic trajectory and foreign economic cooperation over the medium to long term. Against the backdrop of escalating geopolitical tensions and global recession, these challenges could weaken China’s adaptive capacity and further amplify the negative impacts of deglobalisation. Second, while the study included sectoral policies^{49,55} related to education, health, energy, water, and land, more specific policies were not analysed in the study, including fertilizer consumption, dietary changes, carbon pricing policies, postponed retirement, as well as monetary and fiscal policies aimed at directly promoting economic development. Although these specific policies might yield influence on disruptors, integrating these policies needs a model capable of simulating the economy with enhanced precision and depth. Such complexity extends beyond the scope of the iSDG model, which is designed to capture broader interactions within the entire system. Future improvements in modelling capabilities could allow for a more detailed exploration of how these policies affect sustainability.”

Moreover, we added Supplementary Table 6 in the revised Supplementary Information to show that the GDP projections cover a wide range of economic growth

rate in our iSDG-China model.

Supplementary Table 6. The long-term GDP projection of the iSDG-China model and the comparison against the GDP projections of several major models.

Models	Units	GDP Projections in 2050	Average annual GDP growth rate (%), 2020-2050
iSDG-China	Trillion 2015 US\$	20.7–57.8	1.1–4.6
Jing et al. ⁶¹	Trillion 2010 RMB	180–278	2.6–4.2
Jiang et al. ¹⁰⁹	Trillion 2015 US\$	26.0–58.8	1.9–4.7
IIASA	Trillion 2005 US\$ PPP	39.2–62.8	1.7–3.2
OECD	Trillion 2005 US\$ PPP	39.1–86.2	2.1–4.8
PIK	Trillion 2005 US\$ PPP	42.7–64.4	2.3–3.6

Comment 4: Finally, it is not very clear to me how ‘economic growth’ projections are integrated in the medium/long term simulations. Historical trends (as presented in “Supplementary Figures”) may not be a good guide for global or Chinese economic growth dynamics in the next 7-20 years.

Authors’ response: We apologize for the lack of clarity. To address the comment, we added the details to show how to calculate China’s economic output within the iSDG model in the revised Supplementary Information.

“In brief, total economic output calculation is defined as the sum of the economic output of agriculture, industry and services sectors, formulated by Equation (1), using an extended Cobb-Douglas production function for each sector. A detailed description of economic output calculation (e.g., major assumptions, exogenous input variables, initialization variables and source literature) is available in the iSDG model documentation⁵⁶.”

$$GDP = EO_{agr} + EO_{ind} + EO_{ser} \quad \text{Equation (1)}$$

where GDP denotes gross domestic product; and EO_{agr} , EO_{ind} , and EO_{ser} denote the economic output of agriculture, industry and services sectors, respectively.

The agriculture production is formulated as follows.

$$EO_{agr} = EO_{crops} + EO_{livestock} + EO_{forestry} + EO_{fishery} \quad \text{Equation (2)}$$

where EO_{crops} , $EO_{livestock}$, $EO_{forestry}$ and $EO_{fishery}$ denote crops production, livestock production, forestry production and fishery production, respectively. Production factors of the agriculture production include land, capital and labour, and factor productivity depends on several other drivers, including: infrastructure, education, health, governance, access to electricity, macroeconomic stability, female participation in the workforce, openness to trade, climate change, energy prices and public expenditure in agriculture sector.

The industry production is formulated as follows.

$$EO_{ind} = TFP_{ind} L_{ind}^{1-\beta} K_{ind}^{\beta} \quad \text{Equation (3)}$$

where L_{ind} and K_{ind} are the labour and capital of the industry sector; and β denotes the elasticity coefficient of capital-output of the industry sector; and TFP_{ind} is total factor productivity for the industry sector, which depends on several drivers including: infrastructure, education, health, governance, access to electricity, macroeconomic stability, female participation in the workforce, openness to trade, climate change, and energy prices.

The services production is formulated as follows.

$$EO_{ser} = TFP_{ser} L_{ser}^{1-\gamma} K_{ser}^{\gamma} \quad \text{Equation (4)}$$

where L_{ser} and K_{ser} are the labour and capital of the services sector; and γ denotes the elasticity coefficient of capital-output of the services sector; and TFP_{ser} is total factor productivity for the services sector, which depends on several drivers including: infrastructure, education, health, governance, access to electricity, macroeconomic stability, female participation in the workforce, openness to trade, climate change, and energy prices.

A detailed calculation of the agriculture, industry and services production is available in the iSDG model documentation⁵⁶.

We agree with you that it is impossible to accurately predict future economic growth based on historical trends. Instead, we added a comparison of GDP projections between our model and other models (see Supplementary Table 6 of the revised Supplementary Information), which shows that the scenarios we identified in the manuscript cover a very wide range of economic growth, particularly for pessimistic economic outlooks.

Supplementary Table 6. The long-term GDP projection of the iSDG-China model and the comparison against the GDP projections of several major models.

Models	Units	GDP Projections in 2050	Average annual GDP growth rate (%), 2020-2050
iSDG-China	Trillion 2015 US\$	20.7–57.8	1.1–4.6
Jing et al. ⁶¹	Trillion 2010 RMB	180–278	2.6–4.2
Jiang et al. ¹⁰⁹	Trillion 2015 US\$	26.0–58.8	1.9–4.7
IIASA	Trillion 2005 US\$ PPP	39.2–62.8	1.7–3.2
OECD	Trillion 2005 US\$ PPP	39.1–86.2	2.1–4.8
PIK	Trillion 2005 US\$ PPP	42.7–64.4	2.3–3.6

Comment 5: Brief comments: i. The difference of the impact between ‘ageing only’ (-39.7%) and ‘all disruptors’ (50%) scenarios on ‘government revenue’ seems rather small?

Authors’ response: Thank you for your insightful query. First, we revised the text

to describe the impacts of all disruptors on government revenue more precisely. Please see Page 6.

“which could undermine the potential of economic growth (SDG 8) and lead to a 52.3% reduction in government revenue”

After investigating this issue, we think that the difference of the impact between “ageing and shrinking population” and “all disruptors” on government revenue is reasonable. When simulating individual disruptors, the two most influential disruptors affecting government revenue were ageing and shrinking population (-39.7%) and deglobalization (-25.3%), while the impacts of other disruptors were relatively small (from -0.5% to 2.9%). The interactions between ageing and shrinking population and deglobalization cannot result in fully additive consequences due to their overlapping impacts on the economy.

Comment 6: ii. The differential impact that disruptors (e.g. ‘ageing’) has on environmental variables (biodiversity vs. CC) could be discussed a bit more (see also the respective literature on economic/financial crises and economic slowdowns).

Authors’ response: Following your comment, we added the relevant literature on the differential environmental impacts of ageing, and also added the following text to clarify this issue in the discussion section. Please see Page 10.

“In line with previous research efforts^{37–39} attempting to explore the implications of ageing on China, we found ageing and shrinking population posed significant challenges to the labour force vital for economic production, and placed unprecedented stress on the healthcare system, which would collectively reshape the trajectory of China’s long-term economic development, thereby severely affecting the progress of good health and well-being (SDG 3) and decent work and economic growth (SDG 8) (Fig. 2). Ageing and shrinking population also contributed to reductions in material consumption and emissions, affirming the findings from previous studies^{40,41}. However, we found that it failed to deliver sustained and widespread environmental benefits, as evidenced by the continued poor performance in biodiversity (SDGs 14 and 15) under the assumption that government expenditure followed the baseline trend (Supplementary Table 3). In fact, the rising financial burden of ageing may force a reassessment of public spending priorities, potentially leading to reduced investments in land management, biodiversity conservation, and energy transition. Moreover, the preference of older adults for conventional products and services may weaken market demand for innovation and emerging technologies^{42,43}, potentially impeding the adoption of sustainable solutions such as the digital revolution and green consumption, which suggests that the real-world environmental impacts of ageing could be more negative.”

Comment 7: iii. The different ‘levels of ambition’ in policy responses/portfolios are not clearly introduced in the paper. Lines 404-416 refer to three levels: no response, moderate, ambitious. Yet, the paper reports only on ‘no response’ and ‘ambitious’? A clearer presentation of this in the paper would make a difference.

Authors' response: We apologize for the confusion. The details of policy portfolios were presented in Supplementary Table 2. Our study shows the effectiveness of ambitious single-policy portfolios and moderate single-policy portfolios on the overall SDG score.

“Our simulations suggest that, across the scenarios, ambitious single-policy portfolios generated more synergies than moderate single-policy portfolios, with one exception (Supplementary Figs. 5-11). Comparing the below 2°C to the below 1.5°C policy impacts, we found that the additional energy system and carbon dioxide removal costs associated with the more ambitious climate policy brought trade-offs with social and economic sustainability, such as higher government deficits (SDG 17), less funding for healthcare and education (SDGs 3 and 4), ultimately resulting in an inferior overall SDG score (Fig. 4 and Supplementary Fig. 12).”

To further illustrate the performance differences between ambitious single-policy portfolios and moderate single-policy portfolios, we added Supplementary Figs. 5-11 in the revised Supplementary Information.

Comment 8: iv. There seems to be a mismatch between the presentation in Figure 3 and corresponding lines 233-245 (maybe the problem is with the headings ‘no disruptor’, ‘all disruptors’?). Please check. Also, the findings reported here are not very clear to me. Is it that in an ‘all disruptors’ & ‘no response’ scenario the impact on SDG attainment until 2030 is negligible? Is this possible?

Authors' response: Apologies for the confusion. In 2030, the compound effects of all disruptors would result in reductions of 1.3, 2.3, 1.6 and 0.7 points in the overall, economic, social and environmental SDG scores, respectively.

In the revised manuscript, we revised Fig. 3 and its explanations to better describe the impacts of the policy portfolios considering both the states of the world involving no disruptors and all disruptors. If Fig. 3 is still unclear, please let us know.

Figure 3. The overall, economic, social and environmental SDG scores of the policy portfolios under two conditions: the states of the world without disruptors (in light blue) and states of the world with all disruptors (in light red) over time. Supplementary Table 4 specifies SDG targets allocated to the economic, social and environmental categories, and the economic, social and environmental SDG scores are the mean of SDG targets for corresponding categories, respectively. The red line indicates the score of the baseline policy portfolio (no response), the blue line represents the score of the best policy portfolio (Edu+, UHC+, Below 2°C, Bio+ and Water+). The dark grey shading indicates the score range of single-policy portfolios over time. All shading (light grey shading and dark grey shading) indicates the score range of all policy portfolios over time. The numbers in coloured rectangles indicate the lower (in red) and upper bounds (in blue) of the score range of differences as compared against the baseline policy portfolio in the overall, economic, social and environmental SDG scores by 2030 and 2050. The boxplots indicate score distributions of the policy portfolios under the states of the world involving no disruptors (in light blue) and all disruptors (in light red) in 2030, 2040 and 2050, the red points indicate the scores of the baseline policy portfolio, and the blue triangles indicate the scores of the best policy portfolio.

Comment 9: v. In Figure 2: why there are no data for some variables?

Authors' response: Thank you for your careful examination of our manuscript. To ensure that Fig. 2 remains clear and interpretable, cells within Fig. 2 were left blank where the corresponding values were small, specifically falling within the range of -0.1 to 0.1. And we clarified this issue in the explanations of Fig. 2:

“Cells are blank where the differences were small (i.e., between -0.1 and 0.1).”

Comment 10: vi. In ‘Extended Data Figure 6’. Check the data for Government revenue. Why does it increase with Climate Change? Am I misreading this?

Authors’ response: Thank you for your insightful query. We carefully checked the data for government revenue, and confirmed that the model outputs. Moderate increases in temperature have been shown, in some scenarios, to be correlated with increased economic productivity (Burke et al., 2015; Liu et al., 2021), thus boosting economic development and government revenue through taxation.

Therefore, as indicated in “Extended Data Figure 6” (Supplementary Fig. 2 in the revised Supplementary Information), there was an approximate increase of 2.8% (241 vs. 247.9 trillion Yuan) in government revenue associated with climate change by 2050.

In “Calculation of economic output” sub-section of the revised Supplementary Information, we added explanations about the economic impacts of heat-related changes due to climate change on labour productivity within China.

“Production factors of the agriculture production include land, capital and labour, and factor productivity depends on several other drivers, including: infrastructure, education, health, governance, access to electricity, macroeconomic stability, female participation in the workforce, openness to trade, climate change, energy prices and public expenditure in agriculture sector.”

“ TFP_{ind} is total factor productivity for the industry sector, which depends on several drivers including: infrastructure, education, health, governance, access to electricity, macroeconomic stability, female participation in the workforce, openness to trade, climate change, and energy prices.”

“ TFP_{ser} is total factor productivity for the services sector, which depends on several drivers including: infrastructure, education, health, governance, access to electricity, macroeconomic stability, female participation in the workforce, openness to trade, climate change, and energy prices.”

References:

1. Burke, M., Hsiang, S. M. & Miguel, E. Global non-linear effect of temperature on economic production. *Nature* **527**, 235–239 (2015).
2. Liu, Y. et al. Assessment of the Regional and Sectoral Economic Impacts of Heat-Related Changes in Labor Productivity Under Climate Change in China. *Earth’s Future* **9**, e2021EF002028 (2021).

Thank you again for your constructive comments and support! Your comments and recommendations have aided us in strengthening this work.

Reviewers' Comments:

Reviewer #1:

Remarks to the Author:

I am glad the authors have responded well to my earlier comments and concerns. The paper has been revised well.

Reviewer #2:

Remarks to the Author:

The authors have carefully and effectively addressed all my concerns.

-The Discussion section is adequately strengthened and makes clearer the main messages/implications of the research/findings. Therefore, the main weakness of the original paper has been adequately addressed.

-The presentation of the methodology is clearer and more effective throughout the paper. The new brief section on weaknesses is also a needed and valuable addition.

-All other queries that I raised were effectively addressed. Furthermore, the additions made throughout the text (and supplementary material) have significantly strengthened the paper.

-The level of engagement with relevant existing literature remains modest, but this does not limit the original contribution and impact of the paper.

Taking into consideration all the above, I would recommend the publication of the paper.

PS. The trade-off between CC action and socio-economic development identified in lines 224-230 has potentially significant policy-making implications. If any of the paper's findings allow the authors to comment on how this trade-off can be addressed, or is counter-balanced, through other broader (policy) synergies, I would consider adding 1-2 lines in the text to this effect. Yet, I am happy to leave this with the authors, and do not request this as a needed revision.

Reviewer #1's Comments:

I am glad the authors have responded well to my earlier comments and concerns. The paper has been revised well.

Authors' response: Thank you very much for your positive comment!

Reviewer #2's Comments:

The authors have carefully and effectively addressed all my concerns.

-The Discussion section is adequately strengthened and makes clearer the main messages/implications of the research/findings. Therefore, the main weakness of the original paper has been adequately addressed.

-The presentation of the methodology is clearer and more effective throughout the paper. The new brief section on weaknesses is also a needed and valuable addition.

-All other queries that I raised were effectively addressed. Furthermore, the additions made throughout the text (and supplementary material) have significantly strengthened the paper.

-The level of engagement with relevant existing literature remains modest, but this does not limit the original contribution and impact of the paper.

Taking into consideration all the above, I would recommend the publication of the paper.

Authors' response: Thank you very much for your positive comment!

PS. The trade-off between CC action and socio-economic development identified in lines 224-230 has potentially significant policy-making implications. If any of the paper's findings allow the authors to comment on how this trade-off can be addressed, or is counter-balanced, through other broader (policy) synergies, I would consider adding 1-2 lines in the text to this effect. Yet, I am happy to leave this with the authors, and do not request this as a needed revision.

Authors' response: We added the text in the main text (see lines 210-212). Thank you!

“Bridging financial gaps can address these trade-offs, e.g., carbon tax with revenue recycling can reduce emissions and improve well-being without undue burden on economic development³³”.